# Assessment of anxiety symptoms with low social support and associated factors among men who have sex with men (MSM): A cross-sectional study

Yi-Wei Zhou[1,2,3], Jun Li[4], Chun-Yan Shan[5], Zu-Mu Zhou[5]*

**1** Business School, University of Shanghai for Science and Technology, Shanghai, China, **2** School of Intelligent Emergency Management, University of Shanghai for Science and Technology, Shanghai, China, **3** Smart Urban Mobility Institute, University of Shanghai for Science and Technology, Shanghai, China, **4** Tuberculosis and AIDS prevention and Control Institute, Wenzhou Center for Disease Control and Prevention, Wenzhou, China, **5** International Collaborations, The Affiliated Kangning Hospital of Wenzhou Medical University Zhejiang Provincial Clinical Research Center for Mental Disorders, Wenzhou, China

* zhouzumu@126.com

## Abstract

### Objective

This study aims to examine the levels of anxiety symptoms and perceived social support among the men who have sex with men (MSM) population, to assess the level of both anxiety and low social support, and associated factors in this population.

### Method

The study used an Internet service platform for MSM between March and June 2024. Sociodemographic information, laboratory test data, and scores from the Generalized Anxiety Disorder scale (GAD-7), and Perceived Social Support Scale (PSSS) were collected among men who have sex with men (MSM). Decision tree model and binary logistic regression were used to analyze the factors associated with anxiety with low perceived social support.

### Results

A total of 1070 MSM respondents were recruited, of whom 19.6% had anxiety symptoms, and 12.90% had low social support. The prevalence of anxiety symptoms was significantly higher among individuals with low social support (38.41%) than among those with medium or high social support (16.95%) (P < 0.001). Specifically, 4.95% of all respondents had both anxiety and low social support. Logistic regression analysis showed that employment status (P = 0.028), self-esteem (P < 0.001) and psychological resilience (P < 0.001) were significant factors associated with both anxiety symptoms and low social support in the MSM population. Furthermore, the decision tree

**Data availability statement:** All relevant data are within the paper and its Supporting Information files.

**Funding:** This work was financially supported by 2022 Ministry of Education of China Humanities and Social Science Youth Foundation Project (22YJC790189), Shanghai Key Laboratory of Urban Design and Urban Science, NYU Shanghai Open Topic Grants. (Grant No.2023YWZhou_LOUD), Zhejiang Provincial Clinical Research Center for Mental Disorders Foundation Project, Shanghai University Young Teachers Cultivation and Support Project, and National Social Science Foundation of China Post-funding Project (24FJB002) . The funders had no role in study design, data collection and analysis, decision to publish, or preparation of the manuscript.

**Competing interests:** The authors declare that they have no competing financial interests or personal relationships that could have appeared to influence the work reported in this paper.

model identified self-esteem and psychological resilience as key predictors of both anxiety and low social support in the MSM population (all $P < 0.05$).

## Conclusion

Our study demonstrated that in the MSM population in China's eastern region, the prevalence of both anxiety and low social support was relatively low. Employment status, self-esteem, and psychological resilience were identified as significantly correlated factors for them. To effectively reduce anxiety in this population, interventions should focus on enhancing these factors.

## Introduction

Men who have sex with men (MSM) tend to experience more severe mental health problems than the general population [1,2]. Previous studies have shown that the prevalence of anxiety in the general population was less than 10%. In stark contrast, the prevalence of anxiety in MSM was significantly higher, ranging from 26.4% to 44% [3]. Subsequent studies indicate that the prevalence rate of anxiety and anxiety disorders among the MSM population reaches 32.2%, with the prevalence rate fluctuating between 12.7% and 57.6% [4]. An epidemiological survey in China's western region revealed an anxiety prevalence rate of 21.7% among MSM during the COVID-19 pandemic [3]. These researches highlight the spatiotemporal variation characteristics of the anxiety status of this group. However, research on the anxiety prevalence among MSM in China's eastern region remains limited.

Anxiety in MSM is influenced by multiple factors, including social and cultural factors, psychological and cognitive factors, life stress factors, health – related factors, and internet and social factors. Among these, social and cultural factors (such as lack of social support) and health – related factors (e.g., sexually transmitted diseases) are particularly noteworthy [5–7]. The MSM community often faces societal, family, and self-discrimination, which restricts their access to and perception of social support [5]. Their sexual behavior has largely been unacceptable to the general public, subjecting them to greater social pressure and discrimination. This results in reduced social support and prominent psychosocial problems [6]. MSM with lower social support are more prone to psychological issues [7,8]. This shows a close link between social support and anxiety in the MSM population [9–11], with low social support being significantly and positively associated with increased anxiety risk.

Additionally, psychological resilience is defined as an individual's capacity to adapt and recover in the face of stress, adversity, or trauma [12]. Extensive research has consistently demonstrated a significant negative correlation between psychological resilience and anxiety levels [13,14]. Specifically, individuals with higher psychological resilience tend to experience lower levels of anxiety. This inverse relationship is particularly evident among MSM community. Within this population, those with greater psychological resilience are better equipped to navigate life's challenges and stressors, which in turn helps to mitigate the emergence of anxiety. For instance,

when confronted with social discrimination and psychological stress associated with HIV infection, individuals with higher psychological resilience display significantly lower levels of anxiety. However, it is important to note that the relationship between psychological resilience and anxiety may not be uniform across different genders and cultural contexts, as highlighted by another study [15].

Self-esteem is an individual's overall assessment of themselves, encompassing their perceptions of their abilities, self-worth, and self-image. Studies have consistently demonstrated a negative correlation between self-esteem and anxiety: higher self-esteem is associated with lower anxiety levels [16], particularly within the MSM population [17]. Individuals with high self-esteem are generally more adept at managing life's stressors and challenges, which helps to minimize the onset of anxiety. For instance, when confronted with discrimination, social pressure, or health-related issues such as HIV infection, MSM individuals with high self-esteem tend to exhibit lower levels of anxiety. Conversely, those with low self-esteem are more susceptible to anxiety symptoms, especially among adolescents and young adults. Importantly, self-esteem and anxiety share a bidirectional relationship. Low self-esteem can exacerbate anxiety symptoms, while excessive anxiety can, in turn, undermine an individual's self-esteem [17].

Given the differential impact of regional culture, socioeconomics, and other factors on the mental health of MSM, it is of great significance to conduct epidemiological studies on anxiety among MSM in Eastern China. Our study aims to investigate the levels of anxiety and perceived social support among the MSM population sample, assess the extent of anxiety associated with low perceived social support and its influencing factors, and provide evidence-based recommendations for the government and relevant departments. These recommendations are designed to reduce the likelihood of psychological problems in the MSM population, enhance social support systems, and help prevent and control the occurrence and spread of AIDS/HIV and related diseases.

Numerous models have been developed to identify disease risk factors among the MSM population. When it comes to the methodology for constructing anxiety discriminative models, each model has its own strengths and limitations [18]. For instance, the multivariate logistic regression model used by Miller et al. [19] offers epidemiologically meaningful parameter interpretations and quantifies the significant effects of variables. However, it struggles to capture complex nonlinear relationships due to its linear assumption. In contrast, the Random Forest and its improved algorithm, Balanced Random Forest, applied by Bari et al. [20] can automatically identify nonlinear relationships and interactions between variables. These methods, however, are limited by high model complexity, susceptibility to overfitting, and poor interpretability. In our study, we use both logistic regression and the decision tree algorithm to leverage the strengths of each. This dual – approach strategy offers threefold benefits: First, the decision tree algorithm effectively captures nonlinear relationships and interaction effects between variables, allowing for a comprehensive assessment of each risk factor's contribution to anxiety symptoms and ensuring that complex interactions are included in the discriminative model. Second, logistic regression provides clear and interpretable results regarding risk factors. Third, by comparing the results from both models, we enhance the overall detection accuracy. When applying this combined method to the detection of anxiety among MSM, it meets the requirement in public health research for model interpretability while accommodating the nonlinear characteristics of real-world data. Ultimately, this integrated approach provides a more reliable scientific foundation for formulating precise intervention strategies.

## Subjects and methods

### Participants

This study was a retrospective cross-sectional online survey. The survey was conducted between 15 March and 30 June 2024. This study made use of an Internet service platform for MSM (Sunshine Test) and its staff widely distributed the questionnaire through social platforms such as WeChat groups and QQ groups within the MSM community to ensure more comprehensive and targeted information coverage. The Golden Data website (www.jinshuju.com) was chosen as the questionnaire collection platform. To increase participation, we designed an incentive policy, offering each WeChat or QQ user a 10 RMB gift, like a phone recharge voucher, upon completion as a token of gratitude for their time and contribution. Given the confidentiality of the

MSM community, the survey was conducted anonymously online. To guarantee the authenticity and uniqueness of responses, we restricted each IP address to complete the survey only once in order to prevent duplicate submissions. Respondents were informed of their right to refuse participation at any time and provided informed consent prior to the survey,

Inclusion criteria for this study were as follows: (1) participants who were identified as MSM, including gay men, bisexual men, heterosexual men and transgender men; (2) respondents who were age ≥ 18 years; (3) respondents who were conscious, with basic literacy; (4) individuals who were able to operate mobile phones; (5) participants who gave informed consent to this study and voluntary participation. Exclusion criteria: Entries with missing information, logical inconsistencies, or unusually short completion times (e.g., under 30 seconds) were excluded from the analysis.

## Instruments

The questionnaire, designed by the investigators, consisted of three parts: basic personal information, Generalized Anxiety Disorder Scale (GAD-7), and a set of additional psychological scales, including Perceived Social Support Scale (PSSS), Rosenberg Self-Esteem Scale (RSES) and Connor-Davidson Resilience Scale (CD-RISC). The basic personal information section included age, occupation, marital status, education, monthly income, religious beliefs, employment status, HIV test status, syphilis test status, hepatitis C test status, anti-HIV medication status, sex with men in the past 6 months, sex with women in the past 6 months, HIV disclosure status, and area of residence.

We employed the GAD-7 to assess participants' anxiety levels. The GAD-7 is a widely used self-report scale designed to screen for and assess the severity of generalized anxiety disorder. The scale consists of 7 items, each of which is rated based on the frequency of symptoms experienced over the past two weeks. Scoring criteria: Each item is scored on a 4-point Likert scale, ranging from 0 (not at all) to 3 (nearly every day), with a total score ranging from 0 to 21. Interpretation criteria: 0–4 points: No clinically significant anxiety; 5–9 points: Mild anxiety; 10–14 points: Moderate anxiety; ≥ 15 points: Severe anxiety. A score of ≥10 was considered to be anxiety symptoms [21].

The PSSS is a self-report scale designed to assess the degree of social support that an individual perceives. The scale consists of 12 items divided into three dimensions: family support, friend support, and other support. Scoring and interpretation criteria: Each item is scored on a 7-point Likert scale, ranging from "strongly disagree" (1 point) to "strongly agree" (7 points), with a total score ranging from 12 to 84 points. Interpretation criteria: Scores of 12–36 represent low social support status; 37–60 represent medium social support status; and 6l - 84 represent high social support status [22].

The RSES is used to assess an individual's level of self-esteem. The RSES consists of 10 items, with 5 items being positively worded and 5 items being negatively worded. Each item is scored on a 4-point Likert scale. The total score range for the RSES is from 10 to 40 points. Low self-esteem corresponds to a total score between 10 and 25 points; medium self-esteem corresponds to a total score between 26 and 29 points; and high self-esteem corresponds to a total score between 30 and 40 points [23].

The CD-RISC is a tool developed by American psychologists Connor and Davidson in 2003 to assess an individual's resilience in the face of adversity. The CD-RISC scale includes 25 items, each scored on a 5-point Likert scale. The total score range for the CD-RISC is from 0 to 100 points. Low resilience corresponds to a total score between 0 and 49 points; medium resilience corresponds to a total score between 50 and 74 points; and high resilience corresponds to a total score between 75 and 100 points [24].

The GAD-7, PSSS, RSES, and CD-RISC have all demonstrated robust psychometric properties and have been extensively utilized across diverse populations in China and other countries [25–29]. These scales demonstrate strong reliability and validity, making them suitable for our study [21–29].

## Statistical methods

Binary logistic regression and Chi-squared Automatic Interaction Detector (CHAID) decision tree modeling were used to analyze factors affecting anxiety and social support in the MSM population. The parameters of the CHAID decision tree

model were set as follows: the significance level for node splitting and category merging was set at 0.05. The maximum growth depth of the decision tree was 3 layers. The minimum case number of the parent node and the child node was 100 and 50, respectively.

A node became terminal when its sample size did not meet the minimum requirement, and was not split further. SPSS 26.0 software was used for statistical analysis. Count data were expressed as the number of cases (percentage), and the Chi-square test was used for group comparisons. Statistical inference employed two-sided tests, and differences were considered statistically significant at P<0.05.

Classification model predictions yield four scenarios: True Positives (TP) – correct positive predictions; True Negatives (TN) – correct negative predictions; False Positives (FP) – incorrect positive predictions; False Negatives (FN) – incorrect negative predictions [30]. The performance of classification-based algorithms can be evaluated using accuracy, precision, recall and F1 score [34]. The machine learning performance metrics were calculated as follows: Accuracy = (TP+TN)/ (TP+TN+FP+FN); Precision=TP/ (TP+FP); Recall=TP/ (TP+FN); and F1 score=2/ [(1/recall) + (1/precision)].

### Ethical review

This study was reviewed and approved by the Medical Ethics Committee of The Affiliated Kangning Hospital of Wenzhou Medical University (No 2023014). The research was conducted in line with the Declaration of Helsinki and Good Clinical Practice. All information collected during the survey was kept confidential and not disclosed. A designated person was responsible for storing the data. Respondents were fully aware of how their information would be used. Prior to starting the survey, informed consent was meticulously obtained from every respondent online. It was strictly required that only those who had provided informed consent and were willing to participate were allowed to take part. If informed consent was not acquired from the respondents, or if they expressed a reluctance to engage in the survey, their participation was not permitted.

## Results

### Participant characteristics

A total of 1,113 MSM individuals were recruited for this survey. After excluding 43 respondents due to incomplete questionnaires and logical issues, 1,070 participants met the inclusion criteria, resulting in an effective response rate of 96.1%. The respondents' ages ranged from 18 to 65 years, with a mean age of 28.5±8.5 years. A significant portion, 54.1% (579/1070) were aged 20–29 years. Regarding marital status, 79.3% (848/1070) were unmarried. In terms of education, 41.4% (443/1070) had attained a university education or higher. For monthly income, 37.4% (400/1070) reported an income of ≥6,000 Chinese Yuan (CNY). Concerning religious belief, 23.9% (256/1070) identified as Buddhist. In terms of employment, 71.3% (763/1070) had regular employment. Regarding place of residence, 33.7% (361/1,070) lived in urban areas. When it came to sexual behavior, 62.1% (664/1,070) had sex with men in the past 6 months, while 12.1% (129/1070) had sex with women. In terms of health check-ups, 75.1% (803/1070) had been tested for hepatitis C, 77.0% (824/1070) for syphilis and 87.8% (939/1070) for HIV.

### Univariate analysis of anxiety symptoms and social support in the MSM population

Among the 1070 MSM respondents, 210 cases (19.6%) had anxiety symptoms. Univariate analysis showed statistically significant differences between the anxiety symptoms and the non-anxiety symptoms in the MSM population in terms of age, marital status, income, employment status and receipt of HIV testing (P<0.05). See Table 1.

Further analysis divided the MSM population into two groups based on perceived social support level. Of these,138 participants (12.9%) had low social support, while 932 participants (87.1%) had medium-high social support. Of the 138 individuals with low social support, 53 (38.41%) exhibited anxiety symptoms. Conversely, among the 932 individuals with

**Table 1. Univariate analysis of anxiety symptoms in the MSM population.**

| Variables | Number of respondents | Anxiety symptoms | | | |
| --- | --- | --- | --- | --- | --- |
| | | ≤ 9 | ≥10 | $\chi^2$ | P value |
| Age, years | | | | | |
| ≤19 | 96 | 75 | 21 | 13.928 | 0.008 |
| 20–29 | 579 | 447 | 132 | | |
| 30–39 | 273 | 228 | 45 | | |
| 40–49 | 83 | 73 | 10 | | |
| ≥50 | 39 | 37 | 2 | | |
| Occupation | | | | | |
| Administrative clerks | 115 | 99 | 16 | 6.691 | 0.153 |
| Agriculture, forestry, fisheries and transport staff | 82 | 64 | 18 | | |
| Commercial services | 162 | 138 | 24 | | |
| Professionals | 170 | 136 | 34 | | |
| Others | 541 | 423 | 118 | | |
| Marital status | | | | | |
| Unmarried | 848 | 668 | 180 | 8.104 | 0.017 |
| Married | 183 | 161 | 22 | | |
| Divorced and widowed | 39 | 31 | 8 | | |
| Highest education obtained | | | | | |
| Junior high school or below | 137 | 104 | 33 | 3.273 | 0.351 |
| High School | 189 | 157 | 32 | | |
| Junior college | 301 | 238 | 63 | | |
| University or above | 443 | 361 | 82 | | |
| Monthly income, CNY | | | | | |
| ≤1999 | 163 | 112 | 51 | 27.153 | <0.001 |
| 2000–3999 | 158 | 120 | 38 | | |
| 4000–5999 | 349 | 280 | 69 | | |
| ≥6000 | 400 | 348 | 52 | | |
| Religious belief | | | | | |
| Buddhism | 256 | 206 | 50 | 5.964 | 0.310 |
| Christianity | 83 | 59 | 24 | | |
| Catholicism | 15 | 12 | 3 | | |
| Islam | 6 | 4 | 2 | | |
| Others | 19 | 15 | 4 | | |
| No Religion | 691 | 564 | 127 | | |
| Employment status | | | | | |
| Unemployed | 307 | 220 | 87 | 20.718 | <0.001 |
| Employed | 763 | 640 | 123 | | |
| Place of residence | | | | | |
| Urban area | 361 | 280 | 81 | 2.730 | 0.098 |
| Rural area | 709 | 580 | 129 | | |
| Sexual activity with men in the last 6 months | | | | | |
| No | 406 | 328 | 78 | 0.071 | 0.790 |
| Yes | 664 | 532 | 132 | | |
| Sexual activity with woman in the last 6 months | | | | | |
| No | 941 | 758 | 183 | 0.158 | 0.691 |
| Yes | 129 | 102 | 27 | | |

*(Continued)*

**Table 1.** (Continued)

| Variables | Number of respondents | Anxiety symptoms | | | |
|---|---|---|---|---|---|
| | | ≤ 9 | ≥10 | $\chi^2$ | P value |
| Whether HIV tested | | | | | |
| (−) | 868 | 714 | 154 | 11.096 | 0.011 |
| (+) | 36 | 25 | 11 | | |
| Never tested for HIV | 131 | 97 | 34 | | |
| Unwilling to tell test results | 35 | 24 | 11 | | |
| Whether HCV tested | | | | | |
| (−) | 758 | 618 | 140 | 4.002 | 0.261 |
| (+) | 4 | 4 | 0 | | |
| Never tested for HCV | 267 | 208 | 59 | | |
| Unwilling to tell test results | 41 | 30 | 11 | | |
| Whether syphilis tested | | | | | |
| (−) | 764 | 622 | 142 | 2.018 | 0.569 |
| (+) | 26 | 21 | 5 | | |
| Never tested for syphilis | 246 | 191 | 55 | | |
| Unwilling to tell test results | 34 | 26 | 8 | | |
| Disclosure of HIV test results | | | | | |
| Self-knowledge only | 632 | 510 | 122 | 4.548 | 0.337 |
| Family member knows | 52 | 37 | 15 | | |
| Friends know | 259 | 214 | 45 | | |
| Others | 127 | 99 | 28 | | |
| Whether taking anti-HIV medication | | | | | |
| No | 1031 | 833 | 198 | 3.322 | 0.190 |
| Yes | 39 | 27 | 12 | | |

medium or high social support, 158 (16.95%) showed signs of anxiety. There was a significantly higher rate of anxiety symptoms in the low social support group than in the medium or high social support group ($\chi^2$ = 34.945 P < 0.001). Among all respondents, Overall, 4.95% had both anxiety and low social support. Univariate analysis also showed statistically significant differences (P < 0.05) between the low and medium-high social support groups regarding educational background, receipt of HIV testing, hepatitis C testing, syphilis testing, and use of anti-HIV medications.

## Logistic regression of factors influencing both anxiety symptoms and low social support in the MSM population

Before conducting binary logistic regression, we performed a multicollinearity test, which indicated that all variables had VIF values between 1.1 and 3.1, well below the threshold of 10. Subsequently, we incorporated variables that were significant in the univariate analysis into the logistic regression model. The results revealed that employed MSM individuals exhibited an odds ratio (OR) of 0.405 (95% CI: 0.181–0.906) for experiencing anxiety symptoms with low social support relative to their unemployed counterparts (P < 0.05). Additionally, self-esteem displayed a regression coefficient of −0.227, signifying that higher self-esteem was linked to a lower likelihood of anxiety symptoms under conditions of low social support among MSM individuals. Specifically, the OR for self-esteem was 0.797 (95% CI: 0.732–0.867), implying that each one-unit increase in self-esteem corresponds to a 20.3% reduction in the relative risk of developing anxiety symptoms. In a similar vein, psychological resilience presented a regression coefficient of −0.048, indicating that enhanced psychological resilience was associated with a lower probability of anxiety symptoms in the context of low social support. The OR for psychological resilience stood at 0.953 (95% CI: 0.929–0.978), suggesting that each one-unit increase in psychological

resilience is tied to a 4.7% decrease in the relative risk of anxiety symptoms. Overall, these findings highlight that employment status, self-esteem, and psychological resilience are pivotal factors related to anxiety symptoms in the MSM population with low social support. See Table 2 for details.

### CHAID decision tree modeling of factors influencing anxiety symptoms with low social support in the MSM population

The CHAID model showed that self-esteem and psychological resilience were the significant factors associated with anxiety with low social support among MSM, with self-esteem emerging as the primary predictor at the first level of the decision tree. Specifically, among MSM individuals, the prevalence of anxiety with low social support was 24.0%, 6.5%, and 1.1% for those with self-esteem scores of ≤ 21, 21–25, and >25 points, respectively, indicating a clear downward trend in anxiety cases as self-esteem scores increased ($\chi^2 = 115.735$, $P < 0.001$). At the second level of the decision tree, for individuals with self-esteem scores of 21–25, psychological resilience was the next key factor. Among the 248 individuals, the prevalence of both anxiety and low social support was 10.9% for those with psychological resilience scores ≥ 47, compared to 0.9% for those with scores < 47 ($\chi^2 = 10.062$, $P < 0.05$). Overall, the decision tree analysis highlighted the critical roles of self-esteem and psychological resilience in influencing anxiety with low social support in the MSM population, findings consistent with our logistic regression results. See Fig 1.

The decision tree model in this study demonstrated strong performance with an accuracy of 96.17%. It had a precision of 57.69% and a recall of 84.91%, indicating good balance between these two metrics. The specificity was high at 96.76%, and the Youden's index was 82.67%. The F1 - score, which combines precision and recall, was 0.687.

## Discussions

MSM is an inclusive term used to refer to phenotypic males who have insertive or receptive sex (penile-anal or penile-oral) with other phenotypic males, including people who are transgender or have other gender identities [31]. This population is more susceptible to HIV transmission due to the nature of their sexual activities such as anal sex, and because they often have multiple and irregular sexual partners. Additionally, MSM face significant mental health risk. Contrary to traditional beliefs, this population is not accepted by society and the public at large, which often leads to a variety of psychological problems, such as increased stress, anxiety, depression and suicidal ideation, and the prevalence of these psychological problems is higher among MSM than among heterosexuals [1–2]. Moreover, these individuals tend to have lower levels of social support [29]. However, the changes in anxiety and perceived social support among MSM are influenced by a combination of factors, including individual characteristics, psychological factors, social environment, family situation, and self-identity. Given these pressing concerns, the present study conducted a cross-sectional online survey targeting the MSM population, with the aim of elucidating the prevalence of anxiety and low perceived social support within this group, as well as the prevalence of anxiety concurrent with low perceived social support. Additionally, the study aimed to identify the factors influencing the concurrent occurrence of anxiety and low perceived social support through logistic regression and decision tree models. These efforts are intended to promote mental health, enhance social support, eliminate the negative influencing factors, and ultimately prevent and interrupt the transmission of HIV.

**Table 2. Logistic regression of factors influencing both anxiety symptoms and low social support in the MSM population.**

| Variables | β | S.E. | Wald | Significance | Exp (B) | 95% CI of Exp (B) | |
| --- | --- | --- | --- | --- | --- | --- | --- |
| | | | | | | Lower | Upper |
| Employment | −0.904 | 0.411 | 4.842 | 0.028 | 0.405 | 0.181 | 0.906 |
| Self-esteem | −0.227 | 0.043 | 27.906 | <0.001 | 0.797 | 0.732 | 0.867 |
| psychological resilience | −0.048 | 0.013 | 13.409 | <0.001 | 0.953 | 0.929 | 0.978 |

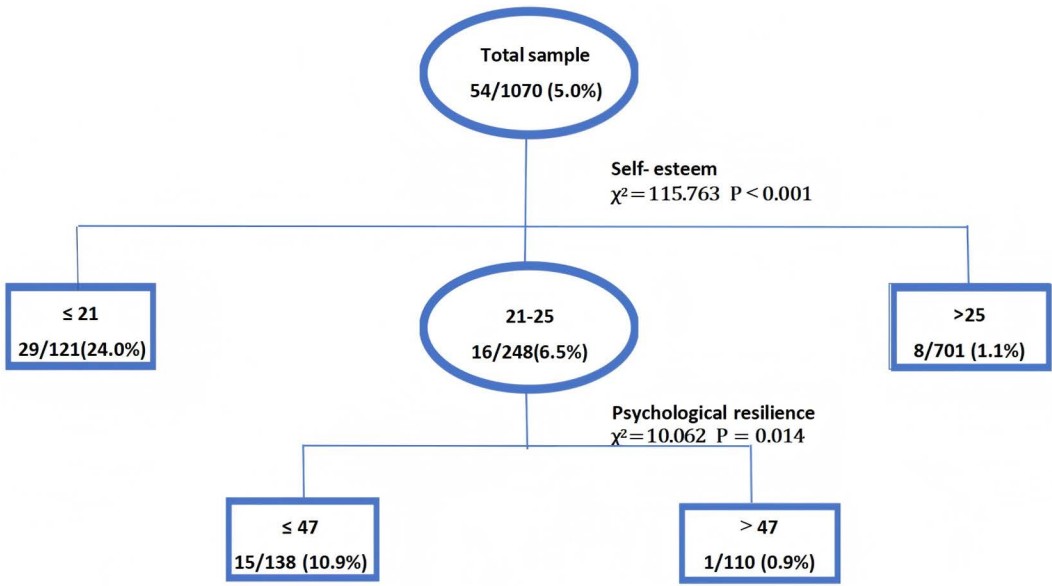

**Fig 1. CHAID decision tree model of the factors related to anxiety symptoms with low social support in the MSM population.**

Several studies have investigated the prevalence of anxiety symptoms among the MSM population [1,4,32,33]. A study in Zambia reported that the prevalence of anxiety among MSM was 45% [1]. Similarly, another survey in Australia showed that the prevalence of anxiety in the HIV-positive MSM population was 36%, which is significantly higher than the 11% prevalence in the general male population in that country [32,33]. However, the present study found that the prevalence of overall anxiety in the MSM population was 19.6%, which was lower than the rates reported in the aforementioned studies [1,32,33]. Overall, the prevalence of anxiety among MSM has been found to vary across different studies, and this variation can be attributed to a multitude of factors, including socio-cultural elements, research methodology, individual characteristics, and social support [4,34].

Social support is a social network that includes three dimensions: family support, friend support, and other support (e.g., social relationships with neighbors, leaders, etc.) [35]. Many studies have shown that social support is closely related to the MSM population [29,36], but there are limited reports on the proportion of low social support within this population. In our study, the prevalence of low social support in the MSM population was 12.9% (138/1070), which was significantly lower than the findings of Lu et al. [37]. This may be related to the different types of perceived social support scales used in the survey and the different criteria for determining low social support.

Several studies have shown that perceived social support is strongly associated with anxiety in the MSM communities [13,38] and that increases in social support significantly reduce anxiety levels [39–41]. In some individuals within the MSM population, anxiety and low social support often coexist. However, according to searches in databases such as PubMed, there have been no reports on the frequency of individuals with anxiety and low social support among the MSM population. Our survey shows that the proportion of individuals with both anxiety and low social support among the MSM population is 4.95%. This group of people should be given high attention and provided with more social support. Social support is crucial, and good interpersonal relationships and joining support groups can provide emotional support. In addition, anxiety can be reduced through a variety of methods and approaches, such as psychological adjustment, lifestyle changes, and professional intervention. In terms of psychological adjustment, cognitive-behavioral therapy, mindfulness meditation, and relaxation training can help change negative thinking and relieve tension. In terms of lifestyle, regular sleep patterns,

healthy diet, moderate exercise, and developing hobbies can stabilize emotions and divert attention [42]. When necessary, professional help should be sought, such as medication, psychotherapy, and alternative therapies [13].

Our research indicates that unemployment, self-esteem, and psychological resilience are factors related to the co-occurrence of anxiety and low social support among the MSM population [13].

Some studies have shown that unemployment is a risk factor for anxiety [43]. In general, unemployment disrupts daily life and is detrimental to mental health. Due to a lack of a job and income, individuals face worries and concerns about the future, as well as increased pressure to find employment, making them more prone to anxiety [44]. The present study shows that the prevalence of anxiety is higher among the unemployed than among the employed, which is consistent with the aforementioned report [43]. Measures should be taken to increase employment opportunities in the future [43–45].

A cross-sectional study involving over 1,000 Norwegian adolescents revealed a strong negative correlation between self-esteem and anxiety [42]. Similarly, another longitudinal study indicated that low self-esteem predicts increased anxiety levels in the future [17] and raises the risk of anxiety relapse three years later [17]. In our study, within the MSM population, self-esteem is a crucial factor associated with the co-occurrence of anxiety symptoms and low social support, being intricately and significantly linked to anxiety and social support. This finding is consistent with the aforementioned reports [17,42]. Self-esteem indirectly affects anxiety through social support. Individuals with low self-esteem are particularly vulnerable to anxiety, a vulnerability that is significantly amplified by insufficient social support. Moreover, low self-esteem can exacerbate anxiety, which in turn further weakens self-esteem, creating a vicious cycle. Overall, the impact mechanism of self-esteem on anxiety and low social support in the MSM group is complex. Improving self-esteem and enhancing social support are essential measures to improve the mental health of the MSM group. Strong social support networks can empower these individuals to more effectively manage life's challenges, thereby reducing anxiety levels.

Psychological resilience is a significant influencing factor for both anxiety and social support. Sun et al. [13] conducted a study involving 161 HIV/AIDS patients and found that psychological resilience is negatively correlated with anxiety ($r = −0.232$, $P < 0.01$). It also plays a full mediating role between social support and anxiety/depression, contributing 68.42% and 59.34% to the effects of social support and anxiety/depression, respectively. Similarly, Hou et al. [46] reported that psychological resilience is significantly correlated with anxiety ($β = −0.253$, $P < 0.001$). The indirect effect of psychological resilience on anxiety through perceived social support is significant ($ab = −0.147$, 95% CI = −0.199, −0.101), accounting for 57.9% of the total effect. In our study, psychological resilience emerges as a crucial factor associated with both anxiety and low social support among the MSM population sample. These findings align with the reports of the aforementioned researchers [13,46]. Individuals with low psychological resilience often lack effective coping strategies when facing stress, making them more susceptible to anxiety [47]. Moreover, inadequate psychological resilience hampers their ability to establish and maintain social relationships, further diminishing the social support they receive. These data further reveal the impact mechanism of psychological resilience on anxiety and low social support in the MSM population, providing strong support for a deeper understanding of the issue.

In our study, logistic regression analysis can reflect the relationship between anxiety and various variables in the MSM population, yet it can't intuitively show the importance of each influencing factor for anxiety symptoms. The CHAID decision tree model can effectively complement logistic regression by illustrating the importance of each factor for the outcome variable and can be displayed clearly and intuitively in the form of a tree diagram. Moreover, the decision tree algorithm can naturally capture the combinations of features and explore the interactions between these features. Decision tree modeling has been used clinically to discriminate and predict a variety of diseases (e.g., diabetes mellitus, hepatocellular carcinoma, acute myocardial infarction, hemodialysis, mental disorder, etc.) [48–52] and aid treatment decisions [52]. However, after searching several databases like PubMed, no decision tree model combined with logistic regression was found to determine predictors of both anxiety and low social support in the MSM population. In our study, the decision tree's robust feature assessment can accurately identify critical factors for diagnosing individuals with concurrent anxiety and low social support in the MSM population. The results show that self-esteem

and psychological resilience are of great value in the diagnostic process, especially self-esteem. Specifically, MSM individuals with low self-esteem and weak psychological resilience are more prone to anxiety and low social support. The decision tree model clearly shows the key roles of self-esteem and psychological resilience in diagnosis [53]. Combining decision tree algorithm with logistic regression, enables earlier, more comprehensive screening for anxiety and low social support risk factors. This allows targeted interventions to improve social support and prevent or delay anxiety onset and progression.

In our study, the decision tree model demonstrates excellent performance in terms of accuracy (96.17%) and specificity (96.76%), effectively classifying the majority of samples, including the accurate identification of negative cases. With a high sensitivity of 84.91%, it also efficiently detects positive samples. The high Youden's index (82.67%) indicates a good balance between sensitivity and specificity, offering strong diagnostic and classification value. An F1 score of 0.687 shows some balance between precision and sensitivity, but there's room for improvement. However, the model's relatively low precision suggests a higher likelihood of false positives.

This cross-sectional study, conducted between March and June 2024, has limitations in capturing the dynamic nature of anxiety and social support. Since both are influenced by numerous factors, they fluctuate over time. By nature, cross-sectional studies only provide a static data snapshot at a specific time point, making them unsuitable for tracking variable changes over extended periods. To better understand these dynamic factors and reveal their complex temporal patterns, future research should adopt a prospective cohort study design. This approach would enable observation of how anxiety and social support evolve and interact over time, offering insights unattainable through single-time-point studies.

These methodological constraints are further compounded by several specific study limitations: (1) The sample was recruited through an internet service platform, which may not be representative of the entire MSM population; (2) As a cross-sectional survey, the results inherit the inherent shortcomings of this design, particularly the inability to establish causal relationships; (3) The reliance on self-reported data introduces potential recall bias, as participants may inaccurately report their anxiety symptoms or social support levels; (4) The exclusive use of self-reported measures systematically increases the risk of response bias; (5) The narrow temporal focus (March-June 2024) cannot account for potential fluctuations in anxiety and social support across different time periods.

## Conclusion

In this study, we investigated anxiety symptoms and perceived social support in the MSM population, along with their associated factors. Our results showed that out of 1070 participants, 210 (19.6%) exhibited anxiety symptoms. The prevalence of low levels of perceived social support in this population was 12.9%. Notably, 4.95% of individuals displayed both anxiety symptoms and low levels of perceived social support. Logistic regression analysis showed that employment status, self-esteem and psychological resilience were significantly associated with the occurrence of anxiety symptoms with low social support in the MSM population. Specifically, these factors substantially increased the likelihood of experiencing anxiety symptoms in the context of inadequate social support. Subsequent analysis using a decision tree algorithm identified that self-esteem and psychological resilience were key factors related to anxiety symptoms and low perceived social support in the MSM population, with self-esteem having a more substantial impact on the outcome than psychological resilience. Given these findings, it is crucial for the government, related departments and social communities to implement targeted measures for the MSM group. These measures should focus on enhancing employment opportunities, boosting self-esteem, and strengthening psychological resilience. By doing so, we can increase social support and effectively reduce anxiety levels among this population.

## Supporting information

**S1 File. Data on Anxiety with low social support among MSM.**
(XLSX)

## Acknowledgments

We would like to express our heartfelt gratitude to all the staff at The Affiliated Kangning Hospital of Wenzhou Medical University who were involved in the investigation.

## Author contributions

**Conceptualization:** Zu-Mu Zhou, Yi-Wei Zhou.

**Formal analysis:** Jun Li.

**Project administration:** Chun-Yan Shan.

**Writing – original draft:** Yi-Wei Zhou.

**Writing – review & editing:** Zu-Mu Zhou.

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
