## [Decision Letter · Decision Letter 0]

12 May 2025

Dear Dr. zumu,

Thank you for submitting your manuscript to PLOS ONE. After careful consideration, we feel that it has merit but does not fully meet PLOS ONE’s publication criteria as it currently stands. Therefore, we invite you to submit a revised version of the manuscript that addresses the points raised during the review process.

We look forward to receiving your revised manuscript.

Kind regards,

Yan-Min Xu

Guest Editor

PLOS ONE

Journal Requirements:

[This work was financially supported by 2022 Ministry of Education of China Humanities and Social Science Youth Foundation Project (22YJC790189), Shanghai University Young Teachers Cultivation and Support Project, Shanghai Key Laboratory of Urban Design and Urban Science, NYU Shanghai Open Topic Grants.(Grant No.2023YWZhou_LOUD), and Zhejiang Provincial Clinical Research Center for Mental Disorders Foundation Project.].

4. In the online submission form, you indicated that [The data that support the findings of this study are available from the corresponding author upon reasonable request.].

Additional Editor Comments:

Please revise the current manuscript strictly according to the reviewers' comments. Although the authors examined a potentially important clinical problem in the MSM population, it still has many issues that need to be addressed. Furthermore, the English language of this paper needs professional editing.

Reviewers' comments:

Reviewer's Responses to Questions

**Comments to the Author**

1. Is the manuscript technically sound, and do the data support the conclusions?

Reviewer #1: Partly

Reviewer #2: Partly

Reviewer #3: Partly

2. Has the statistical analysis been performed appropriately and rigorously?

Reviewer #1: No

Reviewer #2: I Don't Know

Reviewer #3: No

3. Have the authors made all data underlying the findings in their manuscript fully available?

Reviewer #1: No

Reviewer #2: No

Reviewer #3: Yes

4. Is the manuscript presented in an intelligible fashion and written in standard English?

Reviewer #1: No

Reviewer #2: Yes

Reviewer #3: No

Reviewer #1: This paper is not standard and rigorous enough for publication.

1-The introduction of the paper was very descriptive, it did not situate the current study in literature or highlight what the gap in the literature is that this study is trying to address. In this sense, authors must provide better connections between variables under analysis.

2-In relation to the contribution of the study to the literature, I did not get a sense from the article that the findings revealed anything other than what we already know. The introduction of the paper was very descriptive, it did not situate the current study in literature or highlight what the gap in the literature is that this study is trying to address.

Another concern is related with the literature gap. It is unclear what the gap that you intend to fill is?;

3-The Subjects and Methods are also too simple. The author should provide detailed descriptions on the scales and statistical analysis methods.

4-The results section did not show any information on the sensitivity and specificity of the models, but in Discussion section the authors said "The sensitivity of the logistic regression model is higher than that of the decision tree model"

5-Overall, the discussion is very descriptive and any statements about the contribution and conclusions of the study are not new. What is the contribution to the literature, what is interesting about your results? The practical implications need to be further explored, as well as the limitations of the study.

Sometimes the discussion gives the impression that it is not very fluid and very descriptive, perhaps because there are few theoretical and empirical links about what is being analysed. In this sense, I suggest to the authors to make the discussion more fluid, organized into sub-topics and highlighting questions such as: why are your results important? What do you bring back to the literature?

Generally, the paper needs to be reviewed for spelling; grammar and punctuation could be improved in terms of the flow of the read.

Reviewer #2: Overall Evaluation:

This manuscript addresses an important and timely public health issue: the mental health and perceived social support of MSM populations, a group that is often underrepresented and stigmatized. The dual use of logistic regression and decision tree analysis provides complementary insights into predictors of anxiety and low social support. The study is commendable in scope, sample size, and relevance. However, several areas require clarification, refinement, and improvement to meet the standards of a high-impact journal. The writing is not very good. But considering that the important of this issue, I'll give you a chance to major revise.

1.The introduction is a bit too brief. Could you please elaborate on the relevant background?

2.Clarity and Grammar

The manuscript is marred by numerous grammatical issues, awkward sentence constructions, and inconsistencies in tense and word choice. Professional language editing is essential before publication.

Examples: “respondents who was conscious” should be “respondents who were conscious”; “If patients was not willing…” should be “If patients were not willing…”

3.Study Design and Methodology

Cross-sectional design limitation: The authors acknowledge this in the limitations section, but stronger emphasis is needed throughout the discussion when making causal statements.

Sampling and bias: The use of a convenience sample via online platforms may introduce selection bias. More discussion on this limitation is warranted, especially regarding generalizability. And, to whom did you distribute the questionnaire links online? Specifically, which groups of people did you send them to? Who filled out these questionnaires were those people from those groups? This question is rather vague. You must clarify this issue.

Age range discrepancy: Inclusion criteria mention ≥18 years, yet the reported ages go down to 14. This contradiction must be clarified and ethically addressed.

4.Statistical Analysis

The rationale for combining logistic regression and CHAID decision trees is conceptually sound but underexplained. The paper would benefit from a brief paragraph comparing the strengths and limitations of each method and why both were used.

The description of the CHAID model lacks clarity and should include model performance metrics (e.g., classification accuracy, sensitivity/specificity).

Multicollinearity: There is no mention of checking for multicollinearity in the logistic regression model, which should be addressed.

5.Results Presentation

Tables and figures need clearer labeling and formatting. Table 1, for instance, is difficult to interpret due to clutter and misalignment.

Figures 1 and 2 (CHAID trees) are not visible in the manuscript; it is crucial these be embedded for peer review with legible text and nodes.

6.Interpretation of Findings

The paper sometimes overstates associations as if they are causal (e.g., "income had an independent influence on anxiety"). This should be rephrased throughout.

The authors suggest that HIV testing directly improves mental health, but this may be more reflective of reverse causality (i.e., those with less anxiety are more willing to get tested). A more nuanced interpretation is needed.

7.Ethical Concerns

The mention of 14-year-old participants contradicts ethical requirements for adult consent and conflicts with the stated inclusion criteria (≥18 years). This could jeopardize the study’s ethical approval status if not clarified or corrected.

8.Literature Integration

While many relevant studies are cited, the manuscript could benefit from stronger integration of recent high-impact studies on MSM mental health and use of machine learning in public health.

Reviewer #3: This study, conducted by Yi-Wei Zhou et al., aimed to assess the anxiety status, social support level, and associated factors among men who have sex with men (MSM) in China. The research utilized an Internet service platform for MSM between March and June 2024, employing decision tree models and binary logistic regression to analyze the factors related to anxiety and perceived social support. The study found that 19.6% of the 1070 MSM respondents had anxiety symptoms, with higher proportions among HIV-positive subjects (30.6%) compared to HIV-negative cases (17.7%). Furthermore, 12.9% of MSM had lower levels of perceived social support. Logistic regression analyses and decision tree model revealed that income and HIV testing, as well as marriage and work status, were independent factors influencing anxiety symptoms in the MSM population, while education level and HIV testing were independent predictors of low social support.

One significant limitation of this study is the potential for selection bias, as the sample was recruited through an Internet service platform, which may not be representative of the entire MSM population. Secondly, the cross-sectional design of the study limits the ability to establish causality between the variables. Thirdly, the use of self-reported data may also introduce recall and other bias, as participants might not accurately report their anxiety symptoms or social support levels. Fourthly, in the title and elsewhere of this paper, the term “anxiety status” is not accurate, which should be anxiety symptoms, because GAD-7 was used. I do not agree to indicate the statistical methods in the title, please clarify the research design of this study here, i.e., a cross-sectional study. Fifthly, in the introduction of the main text, the authors did not review what has been known on the anxiety symptoms in MSM population and what the limitations and current knowledge gaps are. Therefore, rationale for this study is inadequate. The use of decision tree is novel but it seems that the authors knew little about its strengths such as the identification of interactions and ranking the importance of related factors of anxiety. The decision tree figures are also very crude and not standardized. I suggest the authors to review two prior studies in the statistics, and re-analyze and re-draw the figures accordingly (PMID: 36277764 and PMID: 35245996). Sixth, in the methods, please describe GAD-7 and PSSS in detail including their psychometric properties and widely use in Chinese populations (i.e., PMID: 33313137,PMID: 40309591, PMID: 40194485). Lastly, the study's focus on a specific time frame (March to June 2024) may not capture the dynamic nature of anxiety and social support, which could vary across different periods.

**Do you want your identity to be public for this peer review?** For information about this choice, including consent withdrawal, please see our Privacy Policy

Reviewer #1: No

Reviewer #2: No

Reviewer #3: No

---

## [Author Response · Author response to Decision Letter 1]

26 Jun 2025

Response to Review Comments

Reviewer #1: This paper is not standard and rigorous enough for publication.

1-The introduction of the paper was very descriptive, it did not situate the current study in literature or highlight what the gap in the literature is that this study is trying to address. In this sense, authors must provide better connections between variables under analysis.

Response: Thank you for your comments. We've made major revisions to the introduction, better aligning this research with the existing literature, pinpointing the gaps, and enriching the discussion by adding descriptions of key variables and their interrelations. See line 90-157.

2-In relation to the contribution of the study to the literature, I did not get a sense from the article that the findings revealed anything other than what we already know. The introduction of the paper was very descriptive, it did not situate the current study in literature or highlight what the gap in the literature is that this study is trying to address.

Another concern is related with the literature gap. It is unclear what the gap that you intend to fill is?;

Response: Thank you for your insightful comments. We have thoroughly revised the introduction. In this study, we employed a combination of decision tree analysis and logistic regression to investigate the prevalence and influencing factors of both anxiety and low social support among men who have sex with men (MSM) in Eastern China, aiming to address existing research gaps in this area. Please see the revised introduction. See line 131-157.

3-The Subjects and Methods are also too simple. The author should provide detailed descriptions on the scales and statistical analysis methods.

Response: Thank you for your valuable comments and suggestions. We have added detailed information on the study participants (line 166-174,177-178), survey methods, the self-esteem scale, psychological resilience scale (line 187-214), and statistical analysis procedures (line 225-231).

4-The results section did not show any information on the sensitivity and specificity of the models, but in Discussion section the authors said "The sensitivity of the logistic regression model is higher than that of the decision tree model"

Response: Thank you for your comments. You are correct. We have removed this statement from the discussion section.

5-Overall, the discussion is very descriptive and any statements about the contribution and conclusions of the study are not new. What is the contribution to the literature, what is interesting about your results? The practical implications need to be further explored, as well as the limitations of the study.

Sometimes the discussion gives the impression that it is not very fluid and very descriptive, perhaps because there are few theoretical and empirical links about what is being analysed. In this sense, I suggest to the authors to make the discussion more fluid, organized into sub-topics and highlighting questions such as: why are your results important? What do you bring back to the literature?

Generally, the paper needs to be reviewed for spelling; grammar and punctuation could be improved in terms of the flow of the read.

Response: Thank you for your valuable comments and suggestions. We have thoroughly revised the discussion section in accordance with your recommendations. See line 332-341,360-372,382-410,424-452.

Additionally,we have reviewed and revised the manuscript again, with particular attention to grammar, spelling, punctuation, and overall readability.

Reviewer #2: Overall Evaluation:

This manuscript addresses an important and timely public health issue: the mental health and perceived social support of MSM populations, a group that is often underrepresented and stigmatized. The dual use of logistic regression and decision tree analysis provides complementary insights into predictors of anxiety and low social support. The study is commendable in scope, sample size, and relevance. However, several areas require clarification, refinement, and improvement to meet the standards of a high-impact journal. The writing is not very good. But considering that the important of this issue, I'll give you a chance to major revise.

1.The introduction is a bit too brief. Could you please elaborate on the relevant background?

Response: Thank you for your valuable feedback. We have enhanced the introduction by incorporating relevant content and providing more comprehensive contextual background information. See line 91-102,109-133,139-157.

2.Clarity and Grammar

The manuscript is marred by numerous grammatical issues, awkward sentence constructions, and inconsistencies in tense and word choice. Professional language editing is essential before publication.

Examples: “respondents who was conscious” should be “respondents who were conscious”; “If patients was not willing…” should be “If patients were not willing…”

Response: Thank you for your comments and suggestions. We have conducted a thorough review of the grammar, sentence structure, verb tenses, vocabulary, and overall flow of the manuscript, and have made the necessary revisions accordingly. In addition, the sentences “respondents who was conscious...” and “If patients was not willing...” have been corrected as well. See line 173-177.

3.Study Design and Methodology

Cross-sectional design limitation: The authors acknowledge this in the limitations section, but stronger emphasis is needed throughout the discussion when making causal statements.

Sampling and bias: The use of a convenience sample via online platforms may introduce selection bias. More discussion on this limitation is warranted, especially regarding generalizability. And, to whom did you distribute the questionnaire links online? Specifically, which groups of people did you send them to? Who filled out these questionnaires were those people from those groups? This question is rather vague. You must clarify this issue.

Age range discrepancy: Inclusion criteria mention ≥18 years, yet the reported ages go down to 14. This contradiction must be clarified and ethically addressed.

Response: Thank you for your valuable feedback. The limitations stemming from the cross-sectional nature of this study have been thoroughly discussed in the limitations section and further highlighted in the discussion. See line 437-452.

Moreover, the limitations of the web-based survey, such as potential selection bias and limited generalizability, have been addressed in both the Discussion and Limitations sections. In addition to this�the Methods section (Participants subsection) has been enhanced with additional details regarding participant sources and recruitment procedures. See line 437-447.

Regarding the age range discrepancy The eligibility criterion for age is 18 years or older, and the erroneous mention of 14 years has been duly corrected. See line 161-172.

4.Statistical Analysis

The rationale for combining logistic regression and CHAID decision trees is conceptually sound but underexplained. The paper would benefit from a brief paragraph comparing the strengths and limitations of each method and why both were used.

The description of the CHAID model lacks clarity and should include model performance metrics (e.g., classification accuracy, sensitivity/specificity).

Multicollinearity: There is no mention of checking for multicollinearity in the logistic regression model, which should be addressed.

Response: Thank you for your comments. We have provided explanations of the key concepts of logistic regression and CHAID decision trees. Additionally, we have revised and supplemented the discussion regarding their advantages, limitations, application conditions, and rationale for use, see line 141-157,411-416.

Moreover, model performance evaluation metrics have been added to the statistical methods section, see line 225-231. Key performance indicators, including classification accuracy, sensitivity, specificity, F1 score, and Youden’s index, are reported in the results section, see line 318-321, and the model is further evaluated and interpreted in the discussion section, see line 430-436.

With regard to multicollinearity in the logistic regression model, the relevant results have been incorporated into the results section, see line 279-280.

5.Results Presentation

Tables and figures need clearer labeling and formatting. Table 1, for instance, is difficult to interpret due to clutter and misalignment.

Figures 1 and 2 (CHAID trees) are not visible in the manuscript; it is crucial these be embedded for peer review with legible text and nodes.

Response: Thank you for your valuable feedback. In response, we have removed the social support subsection from Table 1 and provided a corresponding textual explanation, see line 264-273. Additionally, Table 2 has been updated with newly calculated data.

With respect to the figures, the original Figures 1 and 2 have been omitted, and a revised Figure 1 has been included.

Moreover, the original Figures 1 and 2 have been removed, and an updated version of Figure 1 has been incorporated into the manuscript.

6.Interpretation of Findings

The paper sometimes overstates associations as if they are causal (e.g., "income had an independent influence on anxiety"). This should be rephrased throughout.

The authors suggest that HIV testing directly improves mental health, but this may be more reflective of reverse causality (i.e., those with less anxiety are more willing to get tested). A more nuanced interpretation is needed.

Response: Thank you for your valuable feedback. We have carefully reviewed the entire text and adjusted the claims that appeared to imply causality without sufficient support. Furthermore, we have revised or omitted discussions concerning the association between HIV surveillance and mental health.

7.Ethical Concerns

The mention of 14-year-old participants contradicts ethical requirements for adult consent and conflicts with the stated inclusion criteria (≥18 years). This could jeopardize the study’s ethical approval status if not clarified or corrected.

Response: Thank you for your comments. We have amended the reported age of one participant who was incorrectly stated as being 14 years old.

8.Literature Integration

While many relevant studies are cited, the manuscript could benefit from stronger integration of recent high-impact studies on MSM mental health and use of machine learning in public health.

Response: Thank you for your valuable review and suggestions. We have removed some of the original references and added high-impact publications focusing on mental health among MSM and the application of machine learning in public health. These updates can be found in the Introduction section (see line 90-130,139-157), the Discussion section (see line 342-410,411-429), and the References section.

Reviewer #3: This study, conducted by Yi-Wei Zhou et al., aimed to assess the anxiety status, social support level, and associated factors among men who have sex with men (MSM) in China. The research utilized an Internet service platform for MSM between March and June 2024, employing decision tree models and binary logistic regression to analyze the factors related to anxiety and perceived social support. The study found that 19.6% of the 1070 MSM respondents had anxiety symptoms, with higher proportions among HIV-positive subjects (30.6%) compared to HIV-negative cases (17.7%). Furthermore, 12.9% of MSM had lower levels of perceived social support. Logistic regression analyses and decision tree model revealed that income and HIV testing, as well as marriage and work status, were independent factors influencing anxiety symptoms in the MSM population, while education level and HIV testing were independent predictors of low social support.

One significant limitation of this study is the potential for selection bias, as the sample was recruited through an Internet service platform, which may not be representative of the entire MSM population. Secondly, the cross-sectional design of the study limits the ability to establish causality between the variables. Thirdly, the use of self-reported data may also introduce recall and other bias, as participants might not accurately report their anxiety symptoms or social support levels. Fourthly, in the title and elsewhere of this paper, the term “anxiety status” is not accurate, which should be anxiety symptoms, because GAD-7 was used. I do not agree to indicate the statistical methods in the title, please clarify the research design of this study here, i.e., a cross-sectional study. Fifthly, in the introduction of the main text, the authors did not review what has been known on the anxiety symptoms in MSM population and what the limitations and current knowledge gaps are. Therefore, rationale for this study is inadequate. The use of decision tree is novel but it seems that the authors knew little about its strengths such as the identification of interactions and ranking the importance of related factors of anxiety. The decision tree figures are also very crude and not standardized. I suggest the authors to review two prior studies in the statistics, and re-analyze and re-draw the figures accordingly (PMID: 36277764 and PMID: 35245996). Sixth, in the methods, please describe GAD-7 and PSSS in detail including their psychometric properties and widely use in Chinese populations (i.e., PMID: 33313137,PMID: 40309591, PMID: 40194485). Lastly, the study's focus on a specific time frame (March to June 2024) may not capture the dynamic nature of anxiety and social support, which could vary across different periods.

Response: We sincerely appreciate the reviewer’s comments and suggestions. We fully agree with your points.

First, regarding the potential selection bias caused by recruiting participants through online platforms; second, the cross-sectional design of this study does not allow for causal inferences; and third, the use of self-reported data may introduce recall bias and other related biases — all three limitations have been clearly addressed in the Study Limitations section. See line 437-444,445-452.

Concerning the fourth point, we have revised the manuscript by replacing the term “anxiety status” with “anxiety symptoms.” In addition, we have removed “logistic regression and decision trees” from the title and replaced them with “a cross-sectional study.”

Regarding the fifth point, we have added descriptions in the Introduction section about the prevalence of anxiety symptoms among MSM populations and their influencing factors in previous studies, as well as the existing gaps and limitations in current knowledge. Additionally, we have elaborated on some advantages of decision tree algorithms, including their ability to identify interaction effects and rank feature importance. See line 139-157. Meanwhile, we have read the literature you recommended and have benefited greatly from them. Accordingly, we have reanalyzed and redrawn the figure.See figure 1.

With regard to the sixth point, we have supplemented the Methods section with information on the psychological measurement properties of the GAD-7, PSSS, resilience, and self-esteem scales. See line 187-214.

Finally, concerning the last comment, we have specifically clarified in the Study Limitations section that the cross-sectional nature of this study limits our ability to fully capture the dynamic characteristics of anxiety symptoms and social support. See line 437-444�451-452.

Thanks to the editors and reviewers again for your time and energy paid to processing the manuscript. We hope the current version of the manuscript is acceptable for publication in this journal.

Sincerely yours,

Zumu Zhou, Professor

The Affiliated Kangning Hospital of Wenzhou Medical University, Wenzhou ,China

---

## [Decision Letter · Decision Letter 1]

11 Jul 2025

Assessment of anxiety symptoms with low social support and associated factors among men who have sex with men (MSM): A cross-sectional study

PONE-D-24-48458R1

Dear Dr. zumu,

We’re pleased to inform you that your manuscript has been judged scientifically suitable for publication and will be formally accepted for publication once it meets all outstanding technical requirements.

Kind regards,

Yan-Min Xu

Guest Editor

PLOS ONE

Additional Editor Comments (optional):

Reviewers' comments:

Reviewer's Responses to Questions

**Comments to the Author**

Reviewer #2: All comments have been addressed

Reviewer #3: All comments have been addressed

2. Is the manuscript technically sound, and do the data support the conclusions?

Reviewer #2: Partly

Reviewer #3: Yes

3. Has the statistical analysis been performed appropriately and rigorously?

Reviewer #2: I Don't Know

Reviewer #3: Yes

4. Have the authors made all data underlying the findings in their manuscript fully available?

Reviewer #2: No

Reviewer #3: Yes

5. Is the manuscript presented in an intelligible fashion and written in standard English?

Reviewer #2: Yes

Reviewer #3: Yes

Reviewer #2: The manuscript is substantially improved. The authors have undertaken a major revision that has successfully addressed all the critical issues regarding methodological clarity, statistical reporting, interpretation, and ethical concerns.

Reviewer #3: The authors have addressed my concerns.

**Do you want your identity to be public for this peer review?** For information about this choice, including consent withdrawal, please see our Privacy Policy

Reviewer #2: No

Reviewer #3: No
